# Left Ventricular Longitudinal Strain Abnormalities in Childhood Exposure to Anthracycline Chemotherapy

**DOI:** 10.3390/children11030378

**Published:** 2024-03-21

**Authors:** Arnaud Rique, Jennifer Cautela, Franck Thuny, Gérard Michel, Caroline Ovaert, Fedoua El Louali

**Affiliations:** 1Cardio-Oncology Department, AP-HM Nord Hospital, 13915 Marseille, France; 2Pediatric and Congenital Cardiology Department, AP-HM Timone Enfants, 13385 Marseille, France; 3Pediatric Oncology Department, AP-HM Timone Enfants, 13385 Marseille, France; 4Marseille Medical Genetics, Inserm UMR 1251, Aix Marseille University, 13331 Marseille, France; 5Laboratory of Biomechanics and Application, UMRT24, Aix Marseille University, 13331 Marseille, France

**Keywords:** cardiotoxicity, strain, anthracycline, physical activity

## Abstract

Current mortality is low in cases of childhood acute leukemia. Dilated cardiomyopathy induced by anthracyclines remains the main cause of morbidity and mortality during mid-term and long-term follow-up. The aim of our study was to analyze the profile of left ventricular alterations in children treated with anthracyclines and to analyze risks and protective factors, including physical activity. Children and young adults with acute leukemia treated with anthracyclines between 2000 and 2018 during childhood were included. The physical activity performed by the patients before and after treatment was quantified in metabolic equivalent tasks, MET.h, per week. An echocardiographic assessment was performed, including strain analysis. Thirty-eight patients with a median age of 5 [3–8] years were included. Dilated cardiomyopathy was diagnosed in 3 patients and longitudinal strain abnormalities were observed in 11 patients (28.9%). Radiotherapy, cumulative anthracycline doses > 240 mg/m^2^, and the practice of physical activity > 14 MET.h per week (after leukemia treatment) were independently associated with strain abnormalities. In multivariate analysis, radiotherapy was significantly associated with an increased risk of LV GLS abnormalities (OR = 1.26 [1.01–1.57], *p* = 0.036), and physical activity > 14 MET.h/week after oncological treatment was significantly associated with a reduction in the risk of LV GLS abnormalities (OR of 0.03 [0.002–0.411], *p* = 0.009). The strain assessment of left ventricular function is an interesting tool for patient follow-up after leukemia treatment. Moderate and steady physical activity seems to be associated with fewer longitudinal strain abnormalities in patients treated with anthracyclines during childhood.

## 1. Introduction

If untreated, acute leukemia remains the most common and lethal type of childhood cancer. Anthracyclines have been used since the 1950s in childhood acute leukemia treatment and have improved prognoses. This success is tempered by the recognition of adverse effects occurring during or after the completion of cancer treatment [1].

Dilated cardiomyopathy induced by anthracyclines remains, however, associated with increased morbidity and mortality during mid- and long-term follow-up [2]. Most studies on childhood leukemia survivors included patients treated many years ago, when anthracyclines and radiation therapy protocols differed from current practices. Despite better knowledge and improved protocols, anthracycline-induced cardiotoxicity remains relevant in the current era [2].

The therapies commonly used for chronic heart failure with reduced left ventricular ejection fraction remain inefficient. In adults, early cardiotoxicity detection and its prompt treatment seem to be crucial for a major improvement in cardiac function [2]. Concerning the younger population, Lipshultz [3] included 18 doxorubicin-treated long-term survivors of childhood cancer and showed that enalapril-induced improvement in LV (left ventricle) structure and function is transient. The primary defect, which is LV wall thinning, continues to deteriorate, and, thus, the short-term improvement was mostly related to lowered diastolic blood pressure. The author also notes that, after 6 years on enalapril, all six patients who had had congestive heart failure at the start of enalapril therapy either died or underwent cardiac transplantation, compared with three of the twelve asymptomatic patients [3]. This underlines the interest of early diagnosis before any symptoms of heart failure.

In addition, the few available pediatric cardio-oncology studies used echocardiographic measurements of left ventricular shortening and ejection fraction. Those parameters may not detect early and subtle myocardial dysfunction and lack the predictive value in identifying children at higher risk of developing symptomatic cardiovascular disease in the future [4]. Other measures to detect early cardiac dysfunction (such as tissue Doppler imaging, myocardial performance index, two-dimensional strain, strain rate, end-systolic wall stress, and velocity of circumferential fiber shortening) may be powerful tools in identifying anthracycline-induced cardiotoxicity in survivors of childhood cancer.

Finally, there are, to date, no specific studies analyzing the impact of physical activity on children treated with anthracyclines. Although there are exercise guidelines for survivors of childhood cancer following treatment and although data are available on exercise in patients with adult cancers, the potential cardioprotective properties of practice in pediatric oncology have received limited attention [4].

Our study aimed to assess anthracycline-induced cardiotoxicity, using echocardiographic strain analysis, and to analyze risk and protective factors, including physical activity.

## 2. Material and Methods

### 2.1. L.E.A. Protocol

L.E.A. (Leucémie Enfant & Adolescent) is a French prospective long-term follow-up program involving all childhood acute leukemia survivors treated in the participating centers. The protocol is organized as a national cohort study entitled the ‘Study of the determinants of health and quality of life of patients after treatment of acute childhood leukemia—Multicenter Prospective Cohort of Childhood and Adolescent Leukemia (L.E.A.)’. L.E.A. was approved by two French ethical agencies: the Persons Protection Committee (CPP) and the French ‘Safety of Medicines and Health Products National Agency’ (ANSM) (code: 2012-A009684-39). The periodicity of the prospective follow-up evaluations, for included patients in the L.E.A. protocol, is organized according to the chronological history of the pathology and the usual patterns of treatment and follow-up implemented in leukemia patients. According to this structured follow-up, L.E.A. protocol included a specialized cardiologic assessment every 2 or 4 years, depending on the previous treatments received.

### 2.2. Study Population

In this monocentric longitudinal cross-sectional study, patients were included in a successive and prospective way between July 2019 and September 2020. All patients were followed by the pediatric oncology unit at Hospital la Timone Enfants of Marseille and were part of the L.E.A. protocol. Echocardiography was performed as part of their usual follow-up scheduled in L.E.A. protocol. To avoid selection bias, no patient was included in an assessment conducted outside of this follow-up. All participants provided informed consent. The inclusion criteria were patients aged <18 years at the time of diagnosis of acute leukemia and treated between 2000 and 2018. The exclusion criteria were patients who refused to participate in the study, patients with a poor echocardiographic window, and patients with Down syndrome.

### 2.3. Data Collection

Upon inclusion, patient demographic parameters were collected. Data extracted from the L.E.A. database included age at diagnosis of leukemia; cumulative dosage of cardiotoxic chemotherapy received, including anthracyclines and mitoxantrone; and the history of whole-body radiation therapy and the cumulative dose if any. The doxorubicin-equivalent anthracycline matching table was used [5]. To quantify physical activity, we used an international MET.h (Metabolic Equivalent of Task) match chart from the compendium of physical activities [6].

### 2.4. Clinical Data

A complete cardio-oncology evaluation was performed upon the latest assessment. The following data were collected:-Personal medical history;-Physical activity practices before leukemia and after remission; evaluated by tracing the number of hours per week of physical activity, including scholarly physical education;-Cardiac symptoms: dyspnea, angina pectoris, palpitations, syncope;-Physical exam: weight, height, blood pressure, and heart rate;-Resting electrocardiogram.

### 2.5. Echocardiographic Data

All echocardiographic data were obtained and analyzed by the same cardiologist (AR), using a PHILIPS EPIQ CVx echocardiography machine (Koninklijke Philips, Amsterdam, The Netherlands). They included:-Linear measurements of left ventricular dimensions were performed, as recommended, in the parasternal long-axis view. The values of LV dimensions (diastolic interventricular septal thickness (IVS), diastolic posterior wall thickness (PWT), end-diastolic diameter (EDD), and end-systolic diameter (ESD)) were obtained perpendicular to the LV long axis and measured at or immediately below the level of the mitral valve leaflet tips. Internal dimensions were obtained with a two-dimensional (2D) echocardiography (2DE)-guided M-mode approach, or from 2D echocardiographic images (to avoid oblique sections of the ventricle).-Left ventricular global and systolic function analysis consisted of the following:Visual assessment of global and regional myocardial function where each segment was analyzed individually in multiple views.Simpson biplane left ventricular ejection fraction (LV EF) where volume measurements were based on tracings of the blood–tissue interface in the apical four- and two-chamber views.Left ventricular global longitudinal strain (LV GLS). LV GLS was performed after optimizing image quality, maximizing frame rate, and minimizing foreshortening in the three standard apical views (apical 2-chamber, apical 4-chamber, and apical 3-chamber views) and averaged. Age-referenced ranges of LV GLS were defined as established by Lévy et al. [7]. Values below a 95% confidence interval were considered abnormal.-Left ventricular diastolic function: early to late diastolic transmitral flow velocity (E/A) ratio (obtained in 4-chamber view with pulsed wave Doppler); early diastolic mitral annular tissue velocity (e′) (obtained in 4-chamber view with tissue Doppler imaging); E/e′ ratio and left atrial volume (LAV) using the Simpson method.-Right ventricular analysis: tricuspid annular plane systolic excursion (TAPSE), S wave velocity to the tricuspid ring, right ventricular shortening fraction (RVSF), right ventricular global longitudinal strain (RV GLS).-Systolic pulmonary artery pressure (PAP) derived from the maximum velocity of tricuspid insufficiency flow.-Visual assessment of right atrial size, tricuspid valve analysis, diameter measurement of inferior vena cava, and visual assessment of its respiratory compliance,-Visual analysis of the pericardium.

All measurements were carried out according to the joint recommendations of the American Echocardiography Society and the European Cardiovascular Imaging Society [8]. Diastolic dysfunction was defined according to the joint recommendations of the American Echocardiography Society and the European Cardiovascular Imaging Society [9].

For the pediatric population, published Z-scores were used for the following parameters: two-dimensional left ventricular dimensions (IVS, PWT, EDD, ESD) [10], LVM indexed to lean mass [11], LAV [12], TAPSE [13], and S wave velocity [14].

Reference values with a published 95% confidence interval were used for the following values: LV GLS [7] and RV GLS [15].

Due to the lack of existing references for other values, adult standards were used on our pediatric patients. For the adult population, the reference values were derived from the joint recommendations of the American Echocardiography Society and the European Cardiovascular Imaging Society [8].

### 2.6. Statistical Analysis

Continuous variables are expressed as a mean (with standard deviation) or a median (with interquartile range: IQR), where appropriate. Discrete or binary variables are presented as a number (percent). A ROC (receiver operating characteristic) curve was used to assess the level of protective effects of physical activity. Two groups (with and without alterations in LV strain) were compared. The χ^2^ test or Fisher’s exact test (if appropriate) was used for categorical variables and the Mann–Whitney U test was used for continuous variables to assess influencing factors. Parameters with statistical significance in univariate analysis (*p* < 0.05) were introduced into multivariate statistical models. Binary logistic regression was performed to identify significant associations. All *p*-values were bilateral, and significance was pronounced for a *p*-value of less than 5%. Statistical analysis was performed using SPSS software, version 22.0 (SPSS, Inc., Chicago, IL, USA).

## 3. Results

### 3.1. Demographics and Treatment

We prospectively included 38 patients (28 males and 10 females) from the L.E.A. cohort, between July 2019 and September 2020. The median age of patients upon inclusion into this study was 14.5 [11–18.5] years. The median age at the beginning of treatment was 5 [3–8] years. There was no CHD or comorbidities. The median cumulative anthracycline dose received was 187 [120–250] mg/m^2^. Seven children (18.4%) received additional full-body radiotherapy with a total irradiation of 12 Gray and stem cell transplants. The population characteristics, including oncology treatments, are summarized in Table 1.

At the latest cardiac assessment, 3/38 patients (7.8%) had dyspnea (2 were evaluated as class II NYHA and 1 class III NYHA). No patients complained of palpitations, syncope, or chest pain. Blood pressure measurements and heart rates were normal relative to age. Cardiovascular examinations were within the normal limits for all patients.

### 3.2. Physical Activity

Fourteen children (14/38, 36.8%) used to practice regular physical activity prior to their diagnosis of leukemia. The median amount was 0 [0–24] MET.h/week. Thirty-six patients (94.7%) practiced regular physical activity after oncology treatment. The median amount was 28 [21.5–42] MET.h/week.

### 3.3. Transthoracic Echocardiography Data

Echocardiographic evaluations were normal in 29/38 patients (76.3%). Left ventricular global longitudinal strain (LV GLS) was reduced in eleven patients (11/38, 28.9%). Among these eleven patients, three (3/38, 7.9%) had an abnormal LVEF (49%, 51%, and 52%) and were under ACE (angiotensin-converting enzyme) inhibitors. Four patients (4/38, 10.5%) had left ventricular diastolic dysfunction, among which three patients (3/4, 75%) had a dilated left atrium.

In the right ventricular assessment, five patients (5/38, 13.2%) had an abnormally reduced TAPSE value. RV GLS was altered in 15 of the 28 analyzed patients (53.6%). No valvular or pericardial abnormalities were observed. Echocardiographic data are summarized in Table 2.

### 3.4. Risk Factors and Protective Factors

For risk factors, we considered two main variables: radiotherapy (yes vs. no) and the cumulative dose of anthracyclines. Patients were categorized depending on the threshold of 240 mg/m^2^ (admitted according to the threshold in adults).

For protective factors, ROC curves were used to identify a protective threshold of physical activity before and after oncological treatment. The ROC curve (Figure 1) for physical activity after leukemia was significant (*p* = 0.004), with an optimal threshold of 14 MET.h/week. No significant link between pre-treatment physical activity and LV GLS impairment could be found.

In univariate analysis, three factors were associated with LV GLS alteration (Table 3):-A cumulative dose of anthracyclines > 240 mg/m^2^;-Radiotherapy;-Regular physical activity (>14 MET.h) after treatment.

The results of binary logistic regression (multivariate analysis) are presented in Table 4. In multivariate analysis, radiotherapy was significantly associated with an increased risk of LV GLS abnormalities with OR = 1.26 [1.01–1.57] and a *p*-value of 0.036; physical activity after oncological treatment was significantly associated with a reduced risk of LV GLS alteration with an OR of 0.03 [0.002–0.411] and a *p*-value of 0.009.

## 4. Discussion

Our study supports the fact that LV GLS is the most sensitive marker of anthracycline-induced cardiotoxicity in children. Indeed, it is well established that LVEF and left ventricular shortening fraction (LVSF) are imperfect parameters to detect early cardiotoxicity. In Moon et al.’s study, despite having a normal fractional shortening, children exposed to anthracyclines have a subclinical derangement of their left ventricular deformation as measured by decreases in strain and strain rate [16]. Indeed, asymptomatic patients with normal LVEF can present a significant impairment in LV GLS, even in the pediatric population [16,17]. LV GLS alterations are known to be associated with abnormal 6 min walk tests as well as impaired quality of life [18]. In our study, LV GLS impairment was diagnosed in 22.5% of the patients, whereas LVEF anomalies were only present in 7% of the patients. LV GLS seems to become altered first and may be a precursor parameter of heart failure. In HER2-positive breast cancer, LV GLS < |19|% was predictive of subsequent cardiotoxicity and a decreased LVEF [19]. Similar results were reported in adolescents receiving anthracycline chemotherapy, among whom a reduction in LV GLS preceded a decrease in LVEF [17]. These data suggest the prognostic value of LV GLS in the assessment of anthracycline cardiotoxicity.

In pediatric patients, there is still no consensus concerning anthracycline cumulative dosage. The thresholds associated with the alteration of LV GLS vary in different studies [17,20,21,22]. For Yu et al., in 134 adult survivors of childhood, adolescent, and young adult cancer, there was no association between cumulative anthracycline dosage and conventional or strain indices of LV systolic function [17]. Meanwhile, Hu et al. showed significant change in LV GLS with different cumulative anthracycline dosages, with 300 mg/m^2^ as the threshold [21]. European adult recommendations [1] define an anthracycline cumulative dose above 240 mg/m^2^ as a higher-risk cumulative dose, requiring a complete cardio-oncological evaluation before continuing treatment. In our patient population, the ‘adult’ threshold (>240 mg/m^2^) was significantly associated with the alteration of LV GLS.

In adults, radiotherapy is considered both an independent risk factor for dilated cardiomyopathy (DCM) as well as a potential factor in the development of anthracycline DCM [3,23]. In 48 patients irradiated during childhood (27 to 51 Gray), Adams et al. described a 42.6% increase in valve disease, an 11.6% increase in systolic dysfunction, and a 25.6% decrease in left ventricular myocardial mass [23]. More recently, Yu et al., pointed to a history of mediastinal radiotherapy in childhood, associated with anthracyclines, as a risk factor for altered LV GLS [17]. In their childhood cancer survivors, LV GLS was worse in the mediastinal radiotherapy group as compared to the non-radiotherapy group, and the prevalence of patients with GLS ≤ |16|% was greater in the radiotherapy group; meanwhile, there was no difference in FS or LVEF [17]. Our study corroborates those findings in children who received 12 Gray of whole-body radiotherapy. The odds ratio was moderate but statistically associated with a reduction in LV GLS and no other cardiac complications related to radiotherapy were observed.

In primary cancer prevention, regular physical activity reduces the risk of cancer by 25% [24]. The practice of moderate physical activity during specific oncological treatment has been shown to improve oncological prognoses in adult cancer [25] and physical activity is now considered as a real adjuvant treatment. In the childhood population, most studies concern adult survivors of childhood cancer. In a multicenter cohort analysis among 15,450 adult survivors of childhood cancer, vigorous exercise in early adulthood and increased exercise over 8 years was associated with a lower risk of mortality [26]. Jones et al. analyzed data from 1187 survivors of Hodgkin lymphoma. They showed that vigorous exercise was associated with a lower risk of CV events in a dose-dependent manner. This association was independent of CV risk profile and treatment [27]. However, those studies referred to survivors of childhood cancer, some of them concerning patients treated with old chemotherapy protocols and who received mediastinal irradiation in the 1970s and 1980s. In addition, the authors mainly focused on cardiovascular events which happened very late in the evolution of dilated cardiomyopathy [27,28].

The originality of our study is in its focus on early markers of cardiomyopathy. Regular physical activity appears to be associated with less LV GLS alteration. The threshold of 14 MET.h/week highlighted by our analysis must be weighted by the small number of patients and by the possible overestimation of physical activity in this pediatric setting. The benefits of physical activity are currently recognized for prevention [29,30] but also treatment of many adults but also pediatric chronic diseases [31,32,33]. International recommendations (World Health Organization) promote the practice of moderate activity of at least 150 min/week (for adults) or vigorous activity of at least 60 min/week (for adolescents) [34].

### Limitations of the Study

Three major limitations must be emphasized. The first concerns the small number of patients with its statistical drawbacks. The second is that this is a longitudinal cross-sectional type of this study. Large prospective studies will be required to confirm our data and build strong recommendations. The third limitation concerns the difficulty of precisely quantifying physical activity, especially in children. This probably led to overestimations in our study. Assessing the number of hours of physical activity solely based on interrogation remains, however, the most frequently used method to assess physical activity. This limitation will, therefore, be shared by other similar studies.

## 5. Conclusions

In our cohort of 38 patients with acute childhood leukemia treated with anthracyclines between 2000 and 2018, LV GLS was more often altered in the case of concomitant radiotherapy or the case of a cumulative dose of anthracyclines > 240 mg/m^2^. There was a statistically significant association between LV GLS alteration and a lack of physical activity. Even if our data need to be confirmed by larger studies, it suggests clinical implications and is oriented towards future directions, nevertheless.

### 5.1. Clinical Implications

In terms of echocardiographic parameters for cardiotoxicity diagnoses, the early and sensitive detection of cardiotoxicity in children treated with anthracyclines has become a necessity. It may serve different purposes, such as selecting patients who would benefit from therapy or identifying those with a very low risk of future heart failure [35]. In cardiology and adult cardio-oncology, the evidence of the added sensitivity and prognostic value of LV GLS over LVEF in predicting severe endpoints is increasing [35] and adult guidelines for echocardiography discourage the use of linear measurements of global LV function, such as shortening fractions [9]. Given the early and sensitive nature of the alteration of the LV GLS and the numerous data on its prognostic value, it seems legitimate to encourage the integration of LV GLS measurements into regular clinical practice for this patient demographic during screening, monitoring, and follow-up.

In terms of the preventive management of cardiotoxicity, in adult survivors of childhood Hodgkin lymphoma, two distinct but related forms of cardiovascular morbidity and mortality are identified: cardiomyopathy associated with heart failure and coronary artery disease (atherosclerosis). Adult studies proved that regular exercise may be as equally effective at decreasing the risk of both maladies [27]. This potential protective role of physical activity against cardiotoxicity, similarly noted in our study, could highlight the importance of incorporating physical activity into survivorship care plans.

### 5.2. Future Directions

Hypothetical therapeutic targets emerge based on this study’s findings. It seems interesting to contrast irradiation as a risk factor with physical activity as a protective factor in their interaction with the cardiovascular system and their relationship with inflammation. This approach was suggested by Jones et al. [27]. Indeed, irradiation, with or without anthracycline chemotherapy, triggers a chronic proinflammatory/pro-reactive oxygen species (ROS) response. The association of this inflammatory situation with secondary adverse lifestyle perturbations such as physical inactivity, decreased VO2peak, and obesity accelerates the development of traditional CV risk factors (hyperlipidemia, hypertension, and type II diabetes). This proinflammatory systemic milieu alters the endothelial lining and arterial wall function and constitutes the primum movens of atherosclerosis. On the other hand, aerobic training, as a part of a physical activity plan, may lower ROS levels and suppress systemic low-grade inflammation through the increased expression of endogenous antioxidant enzyme machinery [27].

These suggested mechanisms behind the protective effects of physical activity against anthracycline-induced cardiotoxicity could open avenues for novel therapeutic targets.

## Figures and Tables

**Figure 1 children-11-00378-f001:**
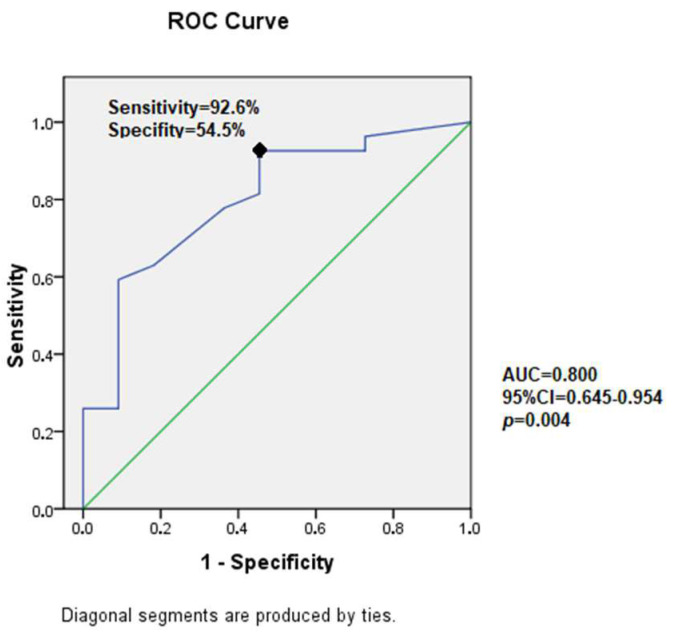
Receiver operating characteristic (ROC) curve analysis for predicting abnormal LV GLS by physical activity expressed in MET.h. Abnormal LV GLS is defined according to reference range values in children as reported by Levy et al.’s meta-analysis [7]. The diagonal line indicates a zero predictive value of the model.

**Table 1 children-11-00378-t001:** Population characteristics.

Population Characteristics	*n* = 38
Sex ratio	28 ♂/10 ♀
Age, median [IQR]	14.5 [11–18.5] years
Age at start of treatment, median [IQR]	5 [3–8] years
Mean cumulative dose (doxorubicin-equivalent), median [IQR]	187 [120–250] mg/m^2^
Total body irradiation 12 Gray, *n* (%)	7/38 (18.4%)

IQR: interquartile range.

**Table 2 children-11-00378-t002:** Echocardiographic data.

Characteristics	*n* = 38
IVS thinning, *n* (%)	2 (5.2%)
ESD enlargement, *n* (%)	1 (2.6%)
EDD enlargement, *n* (%)	0
Abnormal PWT, *n* (%)	0
Decreased LVM, *n* (%)	6 (15.8%)
Decreased LVSF, *n* (%)	3 (7.9%)
Decreased LVEF, *n* (%)	3 (7.9%)
Abnormal LV GLS, *n* (%)	11 (28.9%)
Increased LAV, *n* (%)	3 (7.9%)
LV diastolic dysfunction, *n* (%)	4 (10.5%)
Altered TAPSE, *n* (%)	5 (13.2%)
Altered S wave velocity, *n* (%)	0
Decreased RVSF, *n* (%)	2 (5.3%)
Valvular disease, *n* (%)	0
Pericardial abnormality, *n* (%)	0
LVM indexed, median [IQR]	54.2 g/m^2^ [48.6–65.5]
LVSF, median [IQR]	34% [29–40]
LVEF, mean ± SD	61.2 ± 5.7%
LV GLS, mean ± SD	−20.5 ± 2.8%
RV GLS, mean ± SD	−25.1 ± 4.3%

ESD: end-systolic diameter; EDD: end-diastolic diameter; IQR: interquartile range; LVM: left ventricular mass; LVSF: left ventricular shortening fraction; LVEF: left ventricular ejection fraction; LV GLS: left ventricular global longitudinal strain; LAV: left atrial volume; PWT: posterior wall thickness; RVSF: right ventricular shortening fraction; RV GLS: right ventricular global longitudinal strain; SD: standard deviation; TAPSE: tricuspid annular plane systolic excursion.

**Table 3 children-11-00378-t003:** Predictive factors for left ventricular global longitudinal strain abnormalities.

Parameters	Abnormal LV GLS *n* = 11	Normal LV GLS *n* = 27	*p*
Age in years, mean ± SD	13.0 ± 6.9	15.4 ± 4.8	0.294
Age at leukemia diagnosis in years, mean ± SD	5.86 ± 3.9	6.01 ± 4.1	0.915
Time from treatment to TEE in years, mean ± SD	7.1 ± 6.4	9.4 ± 5.3	0.314
Male gender, *n* (%)	7 (63.6)	21 (77.8)	0.305
Cumulative dose, mean ± SD	242.3 ± 112.9	176.4 ± 104.1	0.074
Cumulative dose >240 mg/m^2^, *n* (%)	6 (54.5)	5 (18.5)	0.036
Radiotherapy, *n* (%)	4 (36.4)	3 (11.1)	0.014
Physical activity before treatment, mean ± SD	7.6 ± 12.6	12.4 ± 18.7	0.374
Physical activity after treatment, mean ± SD	17.1 ± 13.8	36.6 ± 22.8	0.003
Physical activity after treatment >14 MET.h/week, *n* (%)	7 (63.6)	25 (92.6)	0.026

TEE: transthoracic echocardiography; LV GLS: left ventricular global longitudinal strain; SD: standard deviation.

**Table 4 children-11-00378-t004:** Multivariate analysis showing associations between predictive factors and left ventricular global longitudinal strain abnormalities.

Factors	OR	95%	*p*
Cumulative anthracyclines dose > 240 mg/m^2^	4.36	[0.48–39.49]	0.190
Total body irradiation of 12 Gy	1.26	[1.01–1.57]	0.036
Physical activity after treatment > 14 MET.h/week	0.03	[0.002–0.411]	0.009

## Data Availability

The data presented in this study are available on request from the corresponding author due to confidentiality reasons.

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
