# Peer review of "Left Ventricular Longitudinal Strain Abnormalities in Childhood Exposure to Anthracycline Chemotherapy"

_children, 2024, doi:10.3390/children11030378_

Round 1

Reviewer 1 Report

Comments and Suggestions for Authors

The topic of cardiac function after pediatric cancer treatment is certainly clinically relevant. The manuscript is well written and the methods seem correct. However I wonder what the novelty of the presented data is? LV strain abnormalities in pediatric patients after chemotherapy are well described in larger studies. And the correlations with cumulative dose and radiotherapy and the protective effect of exercise are already published in previous articles.

Author Response

Thank you for your review.

The originality of our study is primarily the focus on early markers of cardiomyopathy. LVEF alteration is the visible part of the iceberg and Strain abnormalities may be the immersed part of it. We think that studies with this approach can be useful to promote Strain approche in pediatric cardiac assessment. Secondly, regular physical activity appears to be associated with less LVGLS alteration. Even if it must be weighted by the small number of patients and by the possible overestimation of physical activity in this pediatric setting, the threshold of 14MET.h/week highlighted by our analysis seems to be interesting for further investigation in prospective studies.

Reviewer 2 Report

Comments and Suggestions for Authors

Date: 22/02/2024

COMMENTS

The paper titled "Left ventricle longitudinal strain abnormalities in Pediatric 2 cancer survivors" is a captivating piece of research that falls within the purview of this magazine. Data is available in a limited number of research. A substantial quantity is required to arrive at a sound conclusion. The publishing can be considered once significant revisions have been made to the paper, as outlined below.  

1.      In the Title make the “Pediatric” in small font

2.      In abstract follow the journal guide line and remove the heading  introduction, methods, results from the abstract.

3.      The introduction section need further elaboration and should not contain unnecessary italicized words.

4.      In Method study population must indicate the location of the study and name of the hospital from where data has been taken.

5.      Author must have mentioned the ethical clearance and its number.

6.      Replace the heading Methods with Material and Methods

7.      Give the sub heading number such as 2.1. study population

8.      Clinical and echocardiographic data must be properly summarize of rewrite to avoid the confusion

9.      In results section Demographics and treatment write avoid the multiple paragraph and merge it in one paragaraph from line no 140-147.

10.  In result section give the sub heading number

11.  Merge the line no 188 and 189 togather

12.  Line no 193-195 recheck it is heading sub heading accordingly give the number.

13.   In discussion part remove the all subheading and make a one paragraph for each sub heading.

14.  In Conclusion remove the underline and cross check the font size in whole manuscript it must be as per journal guide line.

15.   Cross check the supplementary material if not provided then remove it.

16.  Line no 276 to313 carefully check and provide the information if not available then delete it.

17.  All references is not as per the journal guide line please correct it.

18.  In Table remove all unnecessary highlighted

Authors must addressed all point to point.

Comments on the Quality of English Language

Date: 22/02/2024

COMMENTS

The paper titled "Left ventricle longitudinal strain abnormalities in Pediatric 2 cancer survivors" is a captivating piece of research that falls within the purview of this magazine. Data is available in a limited number of research. A substantial quantity is required to arrive at a sound conclusion. The publishing can be considered once significant revisions have been made to the paper, as outlined below.  

1.      In the Title make the “Pediatric” in small font

2.      In abstract follow the journal guide line and remove the heading  introduction, methods, results from the abstract.

3.      The introduction section need further elaboration and should not contain unnecessary italicized words.

4.      In Method study population must indicate the location of the study and name of the hospital from where data has been taken.

5.      Author must have mentioned the ethical clearance and its number.

6.      Replace the heading Methods with Material and Methods

7.      Give the sub heading number such as 2.1. study population

8.      Clinical and echocardiographic data must be properly summarize of rewrite to avoid the confusion

9.      In results section Demographics and treatment write avoid the multiple paragraph and merge it in one paragaraph from line no 140-147.

10.  In result section give the sub heading number

11.  Merge the line no 188 and 189 togather

12.  Line no 193-195 recheck it is heading sub heading accordingly give the number.

13.   In discussion part remove the all subheading and make a one paragraph for each sub heading.

14.  In Conclusion remove the underline and cross check the font size in whole manuscript it must be as per journal guide line.

15.   Cross check the supplementary material if not provided then remove it.

16.  Line no 276 to313 carefully check and provide the information if not available then delete it.

17.  All references is not as per the journal guide line please correct it.

18.  In Table remove all unnecessary highlighted

Authors must addressed all point to point.

Author Response

Thank you for your review. 

  1. In the Title make the “Pediatric” in small font

We changed the article title for more clarity

  1. In abstract follow the journal guide line and remove the heading introduction, methods, results from the abstract.

Corrections are made as recommended

  1. The introduction section need further elaboration and should not contain unnecessary italicized words.

Corrections are made as recommended

  1. In Method study population must indicate the location of the study and name of the hospital from where data has been taken.

We added the Information

  1. Author must have mentioned the ethical clearance and its number.

Ethical clearance and number are added

  1. Replace the heading Methods with Material and Methods

Corrections are made as recommended

  1. Give the sub heading number such as 2.1. study population

Corrections are made as recommended

  1. Clinical and echocardiographic data must be properly summarize of rewrite to avoid the confusion

This section has been split in two subheadings to avoid confusion

  1. In results section Demographics and treatment write avoid the multiple paragraph and merge it in one paragaraph from line no 140-147.

Corrections are made as recommended

  1. In result section give the sub heading number

Corrections are made as recommended

  1. Merge the line no 188 and 189 togather

Corrections are made as recommended

  1. Line no 193-195 recheck it is heading sub heading accordingly give the number.

Corrections are made as recommended

  1. In discussion part remove the all subheading and make a one paragraph for each sub heading.

Corrections are made as recommended

  1. In Conclusion remove the underline and cross check the font size in whole manuscript it must be as per journal guide line.

Corrections are made as recommended

  1. Cross check the supplementary material if not provided then remove it.

There is no supplematory material, we remove the paragraph

  1. Line no 276 to313 carefully check and provide the information if not available then delete it.

We added all informations related to the study

  1. All references is not as per the journal guide line please correct it.

We rewrite all refrences as recommended. We cited references 19 and 31 as indicated by the journal on pubmed (as there is no number page)

  1. In Table remove all unnecessary highlighted

Corrections are made as recommended

Reviewer 3 Report

Comments and Suggestions for Authors

General comments: The article presented by Arnaud Rique et al. present a study exploring the association of left ventricle longitudinal strain abnormalities and pediatric in a prospective cohort. The study results are relatively informative.There are some concerns which need to be addressed to improve the manuscript before reconsidered for publication. Specific comments follow:

Overall comments:

1.     The title suggested the manuscript focused on “pediatric cancer survivors”, while the cohorts were children with acute leukemia.

2.     The abstract section should be reorganized. The current format and syntax are confusing.

3.     The introduction did not sufficiently present the objectives of the current study.

4.     There are many ambiguous statements. The Methods-Study population (e.g.: enrollment, exclusion, etc.) needed to be further clarified for the rigor and repeatability. There are ambiguous statements.

5.     Number less than 50 should be avoid using percentage to present their results, because it would cause contain cognitive bias.

6.     Lots of the statements in introduction and discussion section were not well-supported by the current references. Please recheck. Minor language and style revision is recommended. There are various unclear statements due to the grammar and syntax.

7.     Some of the acronyms were not properly sited, which leading it hard to follow.

8.     There were no ethical statements in the current manuscript.

Author Response

Thank you for your review. 

Overall comments:

  1. The title suggested the manuscript focused on “pediatric cancer survivors”, while the cohorts were children with acute leukemia.

We changed this sentence which was indeed not accurate

  1. The abstract section should be reorganized. The current format and syntax are confusing.

We corrected abstract

  1. The introduction did not sufficiently present the objectives of the current study.

We rewrite introduction

  1. There are many ambiguous statements. The Methods-Study population (e.g.: enrollment, exclusion, etc.) needed to be further clarified for the rigor and repeatability. There are ambiguous statements.

We've reworded the paragraph for more clarity

  1. Number less than 50 should be avoid using percentage to present their results, because it would cause contain cognitive bias.

We add ratio to percentage for more precision

  1. Lots of the statements in introduction and discussion section were not well-supported by the current references. Please recheck. Minor language and style revision is recommended. There are various unclear statements due to the grammar and syntax.

We corrected this as recommended

  1. Some of the acronyms were not properly sited, which leading it hard to follow.

We added all cardiologic usual acronyms in abbreviation table. For acronyms of French institutions, we cited them as used in France. Please, tell us if it is clearer.  

  1. There were no ethical statements in the current manuscript.

We added ethical statement with approval number

Round 2

Reviewer 1 Report

Comments and Suggestions for Authors

Questions were answered

Author Response

Thank you 

Reviewer 3 Report

Comments and Suggestions for Authors

- The current vision has improved the quality of the manuscript and covered some of my concerns from my last review.

- However, many ambiguous statements in the Methods section still required extensive improvement. I will continue to list some of the examples, and not limited to:
(1) The study design of this study was to assess the current status of the participants by using a baseline from the past, I would say it is a retrospective cohort or design. If not, please clarify the method to avoid ambiguity.
(2) Also, while the ethics approval is dated 2012, the adult participants in the study were enrolled between July 2019 and September 2020. The timeline is not clearly stated. 
(3) If these patients were since this group of patients enrolled between July 2019 and September 2020 was meant to visit for an advanced cardiology checkup, what measures did the authors take to avoid selection bias in such a design protocol?

- I understand the message the authors trying to deliver from the CMR analysis and the potential results that could validate the hypothesis, however, the authors need to clarify the study design and properly organize the results before further consideration for publication.

Author Response

Dear reviewer

Thank you for your suggestions and remarks, 

(1) The study design of this study was to assess the current status of the participants by using a baseline from the past, I would say it is a retrospective cohort or design. If not, please clarify the method to avoid ambiguity.

Response: Additional data has been added to the methodology section to clarify the situation.

LEA protocol is Prospective long-term follow-up program involving all childhood acute leukemia survivors. Cardiac monitoring is include in the program. This program started in 2012.

We therefore prospectively collected cardiological data during their follow-up appointment organized as part of the LEA protocol. The strain and the entire cardiac exam was done and collected the same day.

(2) Also, while the ethics approval is dated 2012, the adult participants in the study were enrolled between July 2019 and September 2020. The timeline is not clearly stated. 

Response: The first approval for LEA protocol was obteined in 5/12/2012. The approval must be renewed for each request for substantial modification (e.g.: therapeutical protocol, number of centers, etc.).

(3) If these patients were since this group of patients enrolled between July 2019 and September 2020 was meant to visit for an advanced cardiology checkup, what measures did the authors take to avoid selection bias in such a design protocol?

Response: Echocardiography was performed as part of their usual follow-up scheduled in LEA protocol. To avoid selection bias, no patient was included for an assessment done outside of this follow-up.

We have indicated this data in the text.

(4) I understand the message the authors trying to deliver from the CMR analysis and the potential results that could validate the hypothesis, however, the authors need to clarify the study design and properly organize the results before further consideration for publication.

Response: We try to clarify the study design and re organize the results. we hope that will make our work clearer.